# Inactive Proteasomes Routed to Autophagic Turnover Are Confined within the Soluble Fraction of the Cell

**DOI:** 10.3390/biom13010077

**Published:** 2022-12-30

**Authors:** Keren Friedman, Ofri Karmon, Uri Fridman, Yair Goldberg, Ophry Pines, Shay Ben-Aroya

**Affiliations:** 1Faculty of Life Sciences, Bar-Ilan University, Ramat-Gan 52900, Israel; 2NUS-HUJ-CREATE Programme and Department of Microbiology and Immunology, Yong Loo Lin School of Medicine, National University of Singapore, Singapore 138602, Singapore; 3Department of Microbiology and Molecular Genetics, The Institute for Medical Research Israel-Canada (IMRIC), Faculty of Medicine, Hebrew University of Jerusalem, Jerusalem 9112102, Israel

**Keywords:** proteasome, protein quality control, autophagy

## Abstract

Previous studies demonstrated that dysfunctional yeast proteasomes accumulate in the insoluble protein deposit (IPOD), described as the final deposition site for amyloidogenic insoluble proteins and that this compartment also mediates proteasome ubiquitination, a prerequisite for their targeted autophagy (proteaphagy). Here, we examined the solubility state of proteasomes subjected to autophagy as a result of their inactivation, or under nutrient starvation. In both cases, only soluble proteasomes could serve as a substrate to autophagy, suggesting a modified model whereby substrates for proteaphagy are dysfunctional proteasomes in their near-native soluble state, and not as previously believed, those sequestered at the IPOD. Furthermore, the insoluble fraction accumulating in the IPOD represents an alternative pathway, enabling the removal of inactive proteasomes that escaped proteaphagy when the system became saturated. Altogether, we suggest that the relocalization of proteasomes to soluble aggregates represents a general stage of proteasome recycling through autophagy.

## 1. Introduction

Protein misfolding and the formation of toxic insoluble aggregates can have deleterious consequences, which are characteristic of various aggregation syndromes such as Amyotrophic Lateral Sclerosis (ALS) [1], Alzheimer’s and Parkinson’s diseases [2]. To maintain protein homeostasis, cells require intricate protein quality control (PQC) pathways, that mediate protein refolding via molecular chaperones, and target these proteins to proteolytic degradation through the ubiquitin-proteasome system (UPS) and autophagy [3,4,5]. The eukaryotic 26S UPS is a highly conserved 2.5-MD multi-subunit complex capable of catalyzing the degradation of a vast array of intracellular soluble proteins [6,7]. These proteins are usually covalently modified by poly-ubiquitin chains through an enzymatic cascade mediated by several families of enzymes known as E1, E2, and E3 [8,9]. The proteasome assembles from two major subcomplexes, one or two 19S regulatory particles (RP), and a 20S cylindrically shaped core particle (CP), which contains the proteolytic activity [10,11]. Autophagy is uniquely designed to eliminate larger structures, which are encapsulated and delivered in bulk from the cytoplasm to either vacuoles (plants and fungi) or lysosomes (mammals) for breakdown [12,13]. 

Although Ub-mediated proteasomal degradation of many proteins plays a key role in the PQC system, cells also need to dispose of the proteasome components themselves, when they become dysfunctional during their own assembly, an error-prone process that requires the coordinated activity of numerous assembly chaperones [9,11], or as a result of transcriptional and translational failures, genomic mutations, diverse stress conditions, or treatment with proteasome inhibitors, which are widely used to treat various malignancies [14].

As long as functional proteasomes are available, the favored disposal pathway is the degradation of the inactive subunits by the functional 26S complex, similar to other misfolded complexes [15,16]. However, when functional proteasomes become scarce, and a major pathway in the PQC machinery is blocked, alternative pathways are activated [17]. In this case, the yeast small heat shock protein (sHsp), Hsp42, mediates the accumulation of the dysfunctional subunits in cytoprotective cytoplasmic aggregates, which reside in the juxtavacuolar insoluble protein deposit (IPOD) [15,18]. This structure was originally described as the final deposition site for amyloidogenic proteins, including the yeast prions, Rnq1, and Sup35 [19,20]. Notably, the formation of these aggregates was recently identified as a prerequisite for the subsequent clearance of dysfunctional proteasomes by autophagy (termed proteaphagy) [21,22,23]. Following inactivation, 26S proteasomes become extensively modified with Ub. Subsequently, Cue5, which was previously linked to autophagy of polyQ protein aggregates [24], was shown to act as a bridge, linking the ubiquitinated proteasome to the autophagy receptor, Atg8, which coats the autophagosomal membranes [22]. Together, these studies suggested that directing the ubiquitinated dysfunctional proteasomes to the IPOD, next to the vacuole, is a prerequisite step for their subsequent clearance by proteaphagy, allowing Cue5 to deliver the sequestered substrates to the adjacent Atg8, thereby facilitating encapsulation.

Proteasome autophagy can be also triggered by nitrogen deprivation. Like other complexes, proteasome degradation provides a good source of amino acids and nitrogen compounds to replace depleted stores [23,25]. Normally, carbon starvation does not stimulate autophagy; instead, under these conditions proteasomes rapidly re-localize from the nucleus into membrane-free cytoplasmic foci known as proteasome storage granules (PSGs) [18,26]. However, modulating PSG formation by deleting the accessory protein, Blm10, required for their formation, enhances the rate of proteasome degradation, indicating that PSGs help protect proteasomes from autophagic degradation and that PSG assembly and autophagy are mutually exclusive fates of the proteasome [27]. Thus, while both nitrogen and carbon depletion can induce proteasome autophagy, in contrast to dysfunctional proteasomes, this process is not characterized by any type of sequestration to the IPOD, and Hsp42 is not required.

It had been assumed that proteasomes subjected to autophagic degradation are present as insoluble complexes. Here, we proposed that these proteins may be sequestered in a soluble and near-native state. To distinguish these possibilities, we tested the solubility state of dysfunctional proteasomes subjected to autophagy, or under nutrient starvation. We show that under both nutrient starvation and proteasome dysfunction, only soluble proteasomes could serve as a substrate of autophagy. These results suggest a modified model for inactive proteasome recycling through autophagy, in which the substrates for proteaphagy are dysfunctional proteasomes maintained in a near-native soluble state, prior to their sequestration at the IPOD. In addition, we suggest that the insoluble fraction that accumulates in the IPOD may represent a secondary alternative pathway for inactive proteasomes that escaped proteaphagy when the system became overwhelmed. Together, our results indicate that the relocalization of proteasomes to soluble aggregates when they become dysfunctional, or their degradation under nitrogen starvation, represents a general signal for proteasome recycling through autophagy.

## 2. Materials and Methods

### 2.1. Yeast Strains and Growth Conditions

Unless otherwise stated, all the strains used in this study are isogenic to BY4741, BY4742 [28]. The relevant genotypes are presented in Table 1. Deletions and GFP fusions were generated using one-step PCR mediated homologous recombination as was previously described [29]. For all deletions, the selection markers replaced the coding region of the targeted genes. GFP was fused at the 3′ end of the coding region of the targeted genes, by replacement of their stop [29]. A *GAL1* promoter was placed at the N-terminal of *RPN5* and *rpn5*ΔC by replacement of their start codon.

Growth conditions: Yeast cells were grown in synthetic complete medium (SC; 0.17% yeast nitrogen base, 0.5% (NH4)2SO4, and amino acids), supplemented with either 2% glucose (SD), or galactose (SC-GAL). Unless otherwise stated, cells were grown at 30 °C with constant shaking and harvested at the indicated time points by centrifugation. For logarithmic culture, cells were grown for 16–18 h and then back diluted 10x with fresh media and allowed to grow for 2 h. For carbon starvation experiments, cells were grown in a medium containing 2% glucose or galactose to logarithmic phase, collected by centrifugation, washed once, and resuspended in the same medium lacking the carbon source. Alternatively, cells were suspended in the same growth media for at least 4–5 days to deplete the carbon source. For nitrogen starvation, nutrients and ammonium sulfate were omitted from the SD medium.

Standard YEP medium (1% yeast extract, 2% Bacto Peptone) supplemented with, 2% galactose (YP-GAL), or 2% dextrose (YPD) was used for nonselective growth. 2% Bacto Agar was added for solid media. The sporulation medium contained 1% potassium acetate, 0.1% yeast extract, and 0.05% glucose.

### 2.2. Immunological Techniques

GFP liberated from a given protein was monitored as described previously [22]. Total protein extracts from harvested cells were obtained by resuspending cells in 500 μL of yeast lysis buffer (0.2 N NaOH, 1% β-mercaptoethanol), followed by precipitation of proteins with 50 μL of 50% trichloroacetic acid. Proteins were pelleted by centrifugation at 16,000× *g* for 5 min at 4 °C, washed once with 1 mL of ice-cold acetone, and re-suspended in 150 μL SDS-PAGE sample buffer (80 mM Tris-HCl pH 6.8, 10% glycerol, 4% SDS, 4% β-mercaptoethanol, 0.002% bromophenol blue). Samples were then heated at 95 °C for 5 min, and 10–20 μL of each sample was separated by SDS-PAGE and immunoblotted with anti-GFP (1:2500, Roche, 11814460001), and anti-PGK1 (1:2000, Abcam, ab113687).

### 2.3. Detergent Solubility Assay

The Detergent Solubility assay was adapted from [30,31]. Briefly, yeast cells at late logarithmic phase were harvested and lysed using glass beads in 200 μL of lysis buffer (100 mM Tris, pH 7.5, 200 mM NaCl, 1 mM EDTA, 1 mM DTT, 5% glycerol, and 0.5% Triton X-100) for 5 min at 4 °C. Repeated 10sec microcentrifuge pulses cleared the resulting lysates. A 50-μL amount of lysate, representing the “total lysate (T),” was removed and added to 50 μL of SUMEB (8 M urea, 1% SDS, 10 mM 3-(N-morpholino) propanesulfonic acid, pH 6.8, 10 mM EDTA, 0.01% bromphenol blue). The remaining lysate was centrifuged at 17,000× *g* for 15 min. A 100-μL amount of supernatant was added to 100 μL of SUMEB. The pellet was resuspended in 100 μL of lysis buffer plus 100 μL of SUMEB. Proteins were detected by immunoblotting as described.

### 2.4. Microscopy

Cells were observed in a fully automated inverted microscope (Zeiss observer. Z1 Carl Zeiss, Inc.) equipped with an MS-2000 stage (Applied Scientific Instrumentation), a Lambda DG-4 LS 300 W xenon light source (Sutter Instrument), a 63x Oil 1.4 NA Plan-Apochromat objective lens, and a six-position filter cube turret with a GFP filter (excitation, BP470/40; emission, BP525/50). Images were acquired using a CoolSnap HQ2 camera (Roper Scientific). The microscope, camera, and shutters (Uniblitz) were controlled by AxioVision Rel. 4.8.2. Images are a single plane of z-stacks performed using a 0.5 μm step.

## 3. Results

### 3.1. Proteasomes Confined to the Soluble Fraction of the Cells Are Subjected to Autophagy in Response to Nutrient Starvation

Proteasome autophagy can be triggered by nitrogen deprivation or induced under carbon starvation in the presence of Δ*blm10* mutant. Unlike dysfunctional proteasomes, this process is independent of Hsp42 and is not characterized by any type of sequestration to the insoluble fraction that resides at the IPOD [23,25]. Based on these studies, we postulated that proteasomes that are cleared by autophagy in response to nutrient starvation may be present in the soluble fraction of the cell.

Proteasome sequestration and autophagy are typically monitored using GFP-tagged proteasome subunits in their wild-type or mutated forms [11,15,32,32]. To track the CP and RP autophagy, we used a strain with a C-terminus fusion of GFP to the endogenous RP subunit *RPN12*, or to the CP subunit *PRE10*. Since *RPN12* and *PRE10* are essential genes, the successful integration of the GFP clearly indicates that the GFP-tagged version is functional and incorporated into the proteasome. Monitoring the accumulation of a ~25 kDa band that is recognized by the anti-GFP antibody on immunoblots is widely used to detect vacuolar targeting of proteasomes through autophagy. The appearance of this band (hereafter termed “free GFP”) results from the vacuolar cleavage of the linker between GFP and the tagged proteasome subunit, and the folding state of the GFP protein, facilitating its resistance to the vacuolar degradation [21,22,23].

Consistent with previous reports, nitrogen starvation induces the free-GFP fragment associated with the expected autophagy, the presence of which is eliminated in cells deficient in *ATG7*, a core autophagy component required for the autophagosome formation [12,33] (Figure 1a). Furthermore, the deletion of *BLM10* induced the free-GFP fragment under carbon depletion (Figure 1b). To examine the solubility state of proteasomes under these conditions, we adapted the detergent solubility assay, separating total cells extracts (T) to soluble (S) and insoluble pellet (P) fractions of the cell [30,31]. Under both conditions, Rpn12-GFP was mainly confined in the soluble fraction, and notably, the free-GFP was detected only in the soluble fraction of the cells (Figure 1c,d).

Altogether, these results suggest that proteasomes confined at the soluble fraction of the cells under nitrogen and carbon (in Δ*blm10* background) depletion, can serve as substrates for autophagy.

### 3.2. Dysfunctional Proteasomes Confined at the Soluble Fraction of the Cell Can Be Routed to Proteaphagy

We previously found that Hsp42 mediates the accumulation of inactive proteasome subunits at the IPOD [15,16]. Later studies suggested that directing the ubiquitinated inactive proteasome to the IPOD is a prerequisite step for their subsequent clearance by proteaphagy, as the deletion of *HSP42* abolished this process [22]. The IPOD contains mainly irreversibly aggregated amyloidogenic proteins and globally unfolded substrates that form tight protein aggregates [19,20], suggesting that insoluble proteasomes are subjected to proteaphagy. However, based on the results shown in Figure 1c,d, we hypothesized that in analogy to the response to nutrient starvation, dysfunctional proteasome substrates may also be cleared by proteaphagy from the soluble fraction of the cells.

To address this issue, we followed the autophagic cleavage of Rpn12-GFP and Pre10-GFP, under conditions that genetically compromise proteasomes. For the genetic approach, we exploited temperature-sensitive (ts) mutants affecting the RP and CP subunits Rpn5 and Doa5 (*rpn5*Δ*C* and *doa5-ts*, respectively), that were previously used as a research tool for investigating the fate of genetically compromised proteasomes [15,16,18,22,34,35,36], and to track proteaphagy [22,23,27].

Consistent with previous studies, in *rpn5*Δ*C* and in *doa5-ts* cells, free-GFP was induced in cells grown at the semi-restrictive temperature of 34 °C for 8 h (Figure 2a). In all cases, the release of Rpn12, and Pre10 free-GFP was minimal in wt cells and in the control cells deleted in *PEP4*, the vacuolar processing protease [37]. It should be noted that we replaced the Δ*atg7* control used above with Δ*pep4*, as the double Δ*atg7/rpn5*Δ*C* or Δ*atg7/doa5-ts* mutants showed synthetic growth defect.

Next, we used the detergent solubility assay, to determine the solubility state of dysfunctional proteasomes in *RPN12*-GFP *rpn5*Δ*C* cells grown at the 34 °C. The results show that Rpn12-GFP was confined mainly in the soluble fraction after 6 h (83% of the total cell lysates) (Figure 2b-left), with increased enrichment in the insoluble fraction (88% of the total cell lysates) after longer incubation (8 h, Figure 2b-right). Notably, in both cases, the free-GFP was detected only in the soluble fraction of the cells.

These results suggest that inactive proteasomes confined at the soluble fraction of the cells can serve as the substrates for proteaphagy. However, it is still possible that uptake of insoluble proteasome inclusions from the IPOD into the vacuole might lead to solubilization of the aggregates due to the very low pH environment, or that the free-GFP represents the release of cleaved, soluble GFP from insoluble proteasomal particles. To address this issue, we used the conditions described above to induce genetic (*rpn5*Δ*C* and *doa5* ts mutants) proteasome inactivation, for a limited time (3 h), to capture soluble proteasomes, before their relocation to the insoluble fraction. A “free-GFP” band was detected, even when proteasomes were enriched solely in the soluble fraction (Figure 2c), indicating that dysfunctional proteasomes undergo proteaphagy from the soluble fraction.

### 3.3. Dysfunctional Insoluble Proteasomes That Accumulate in the IPOD Represent an Alternative Pathway for Proteasomes That Escape Proteaphagy

It was proposed previously that the IPOD, containing irreversibly misfolded insoluble aggregates, represents an intermediate compartment before autophagic clearance. However, our findings show that the substrates for proteaphagy are dysfunctional proteasomes in their soluble state, while prolonged proteasome inactivation results in dysfunctional proteasome enrichment of the insoluble fraction (Figure 2b-right). Hence, it is possible that the fraction present in the IPOD represents proteins directed to an alternative pathway for disposal of dysfunctional proteasomes that escaped proteaphagy when the system becomes overloaded.

To test this possibility, we saturated the proteophagic machinery using a galactose inducible promoter (*GAL1*), that constitutively expresses *rpn5*Δ*C* (GFP*-rpn5*Δ*C*). When overproducing this mutant, GFP*-rpn5*Δ*C* was predominantly insoluble (Figure 3a, right) and, as indicated by fluorescence microscopy, was completely confined as a single focused juxtavacuolar site that co-localizes with Hsp42, that serves as an IPOD marker [16,38,39,40] (Figure 3c, top). As shown in Figure 3b, no free-GFP was detected in this case, most probably because insoluble proteasomes are not subjected to proteaphagy. By using *rpn5*Δ*C* strain in which the expression of *ATG8* is controlled through a copper (CuSo4)-inducible promoter [41] (p*CUP1*-9xMyc*-ATG8*), we show that when compared to the wt *RPN5* background, Atg8 becomes conjugated to the lipid phosphatidylethanolamine (Atg8-PE) of the autophagosomal membrane (Figure 3d) which indicates autophagosome formation [12,42]. These results exclude the possibility that *rpn5*Δ*C* mutant impairs autophagy. Next, we overproduced *rpn5-1*, another ts mutant of *RPN5* [35] (GFP-*rpn5-1*), that is present both in the soluble cytoplasmic fraction, and in some cases also colocalizes with Hsp42 at the IPOD (Figure 3a left,c bottom). In this case, the free-GFP signal was detected solely at the soluble fraction. This free-GFP signal was eliminated in Δ*atg7* and Δ*hsp42*, indicating that it is the result of autophagic cleavage (Figure 3b).

Altogether, these results are consistent with our hypothesis that insoluble proteasomes are not the substrate for proteaphagy and that the IPOD probably contains the dysfunctional proteasomes that escaped proteaphagy due to the saturation of the autophagy machinery. The observation that *rpn5*ΔC expressed from its endogenous promoter could be detected both at the soluble and insoluble fractions imply that the specific overproduction of this mutant pushes the balance toward the insoluble state and that while proteaphagy of the insoluble fraction may still occur, it is below the detection level of the methods used.

## 4. Discussion

In this study, we provide evidence that the substrates for proteaphagy are dysfunctional proteasomes in their near-native soluble state. We further propose that the insoluble fraction that accumulates in the IPOD, may represent an alternative and secondary pathway for inactive proteasomes that escaped proteaphagy upon saturation of the system.

Hsp42 is a key factor mediating the accumulation of inactive proteasomes at the IPOD [15,22]. The co-localization of inactive proteasomes with the insoluble yeast prion Rnq1 at the IPOD, and the abolishment of proteaphagy in Δ*hsp42* cells, suggested the original model that ubiquitylated proteasomes are directed to the IPOD by Hsp42, as a prerequisite step for autophagic encapsulation and clearance [22]. It is well established that the IPOD contains mainly irreversibly aggregated insoluble proteins [19,20,39,43]; therefore, while not tested experimentally, this model implied that proteasomes subjected to autophagic degradation take the form of insoluble aggregates. In contrast to this initial model, while our results support the role of Hsp42 in mediating proteaphagy, we show here that the substrates for proteaphagy are dysfunctional proteasomes in their near-native soluble state.

We can explain this discrepancy by the well-established role of Hsp42 in orchestrating the regulated coalescence of multiple cytosolic stress-induced aggregates. During unfolding stress, Hsp42 associates with its substrates in a partially unfolded intermediate state, maintaining them in a ready-to-refold conformation close to their native structure, or alternatively, mediates their UPS-mediated degradation [5,44]. Hsp42 co-aggregates with diverse misfolded substrates under different stress conditions, including heat stress [45], proteasome inhibition [15,16,22], cellular quiescence [46], and cellular aging [47]. This co-aggregation is employed to actively control the formation of structures known as CytoQs [5], and to promote their coalescence into a smaller number of assemblies of larger size, until they are sequestered into the IPOD [45,48]. Among other effects, concentrating misfolded proteins at specific deposition sites could facilitate their subsequent refolding by chaperones or clearance by proteolysis. Indeed, a classic example is the aggregates sequestered in CytoQ following heat stress [43,45,49]. In this case, the fate of the sequestered substrates, whether to degradation or refolding, is determined after solubilization by the Hsp70/Hsp100 chaperone system and depends on refolding kinetics of the substrate and its relative affinity for chaperones versus proteases. Based on our results, we propose that dysfunctional proteasomes are subjected to proteaphagy when embedded in CytoQ in their near-native conformation. It is possible that the massive ubiquitination of impaired proteasomes occurs at this stage, allowing Cue5 to bridge between the proteasomes and Atg8-autophagosomes. When the system becomes saturated, Hsp42 aggregase activity takes over, leading to the accumulation of insoluble proteasomes in the IPOD, which is consistent with its role as the final destination for protein aggregates that cannot be disassembled and could thereby become cytotoxic [39,43,49].

Another condition that leads to proteasome autophagy is nitrogen starvation [23,27]. Under such conditions, proteasomes are exported from the nucleus to the cytoplasm, most likely when the holo-complex is dissociated from its CP and RP complexes. Following export, each RP and CP is separately targeted to the Atg8-autophagosomes. In analogy to inactive proteasome sequestration in CytoQ, this process requires the conserved sorting nexin, Snx4, which cooperates with Snx41 and Snx42 to mediate the turnover of proteasomes and several other large multi-subunit complexes by forming cytoplasmic puncta prior to delivery to the vacuole for destruction [25].

Altogether, we suggest that the relocalization of proteasomes to soluble aggregates represents a general stage of proteasome recycling through autophagy, triggered when they become dysfunctional, or under nitrogen starvation. The pathways leading to the formation of these aggregates are most probably distinct since proteasome recycling under nitrogen starvation is not affected by the deletion of *HSP42* or *CUE5* [22]. While it is clear that proteasome aggregation and their distribution between the different deposition sites is not random and is an essential step that enables their proper recycling, further research is required to determine the molecular principles governing sorting to different classes of aggregation sites, under different stress conditions.

## Figures and Tables

**Figure 1 biomolecules-13-00077-f001:**
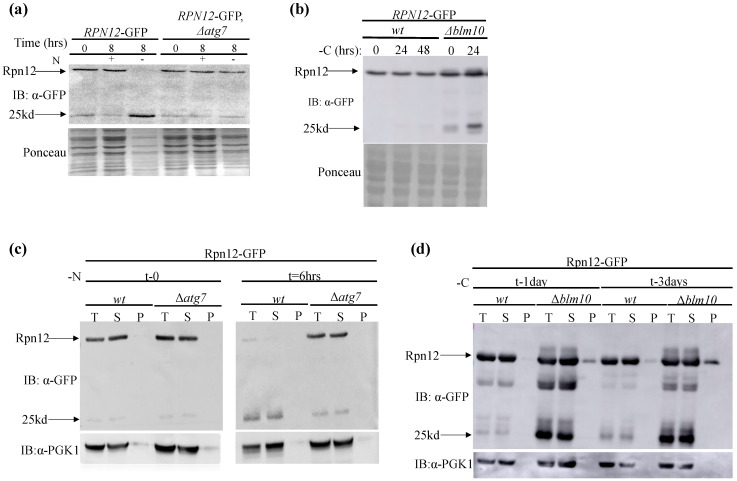
(**a**) Proteasomes in cells grown under nitrogen (**a**) and carbon starvation (in Δ*blm10* cells) (**b**) are subjected to autophagy. (**a**) Logarithmically growing wt cells carrying *RPN12*-GFP and deleted in the control autophagic pathway genes (Δ*atg7*) were grown in a rich medium (t-0). Cells were then washed with sterile water, resuspended in a minimal medium lacking nitrogen (-N), and allowed to grow for 8 h (t-8). (**b**) Similarly, to A, but the indicated logarithmically growing cells (in rich medium) (t-0) were re-suspended in carbon-free medium (-C) to induce carbon starvation for 24, or 48 hrs. Release of free-GFP from the Rpn12-GFP autophagy reporter was assayed by immunoblot analysis of total extracts with anti-GFP antibodies. Total protein Ponceau staining was used as the loading control. (**c**,**d**) Proteasomes confined in the soluble fraction of the cells are subjected to autophagy in response to nitrogen (**c**)**,** and carbon (**d**) starvation. Protein extracts from the cells described in A and B were subjected to detergent solubility assay (see Section 2), and total cell lysate (T), soluble (S), and insoluble pellet fractions (P) were immunoblotted with anti-GFP antibodies to detect Rpn12-GFP distribution and the free-GFP signal. Unless indicated otherwise, in all the free-GFP, and solubility assays, ponceau staining, and anti-Pgk1 served as a loading control, or a to normalize the soluble fraction, respectively. The intact Rpn12-GFP and the free-GFP proteaphagy reporters are indicated by arrows showing the 57 Kd and 25 Kd bands, respectively.

**Figure 2 biomolecules-13-00077-f002:**
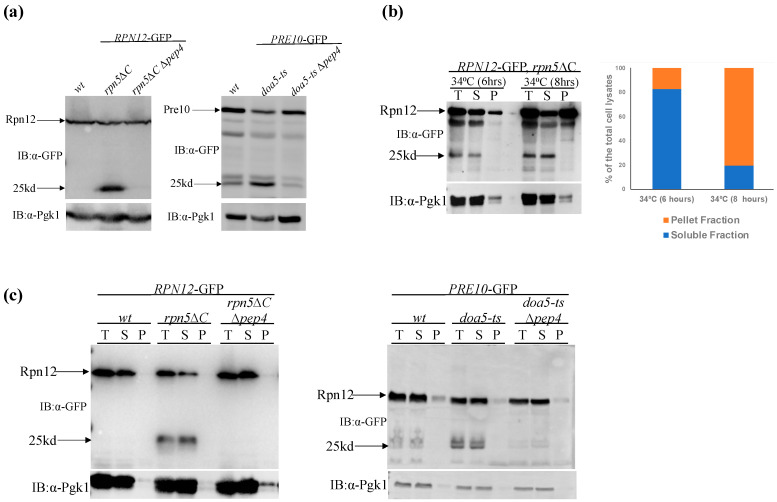
(**a**) Genetically compromised proteasomes are subjected to proteaphagy. Free-GFP assay of cells carrying *rpn5*ΔC or *doa5* temperature-sensitive (ts) mutations, that attenuate the activity of the regulatory particle (RP) subunit Rpn5, or the core-particle (CP) subunit Doa5, respectively, and the 26S proteasome. Cells were grown at the semi-permissive temperature (34 °C) for 8 h. To track CP and RP autophagy, we used the RP subunit *RPN12*-GFP and CP subunits Pre10-GFP. Wt cells, and cells deleted in *PEP4*, the vacuolar processing protease, were used as controls. Pre10-GFP, Rpn12-GFP and the free-GFP proteaphagy reporters are indicated by arrows showing the intact and 25Kd bands, respectively. (**b**) Dysfunctional proteasomes confined at the soluble fraction of the cell can be routed to proteaphagy. (C) *RPN12*-GFP cells, carrying the *rpn5*ΔC ts allele were grown at the restrictive temperature (34 °C) for 6 (left), or 8 h (right), and subjected to the detergent solubility assay as described in Figure 1c,d. The graph quantitates the percentage (average of two independent experiments), of *RPN12*-GFP in the S and P fractions relative to the total cell lysate. (**c**) The indicated strains were grown for a limited time (3 h) at the semi permissive temperature (34 °C) to capture soluble proteasomes before they started to relocate to the insoluble fraction.

**Figure 3 biomolecules-13-00077-f003:**
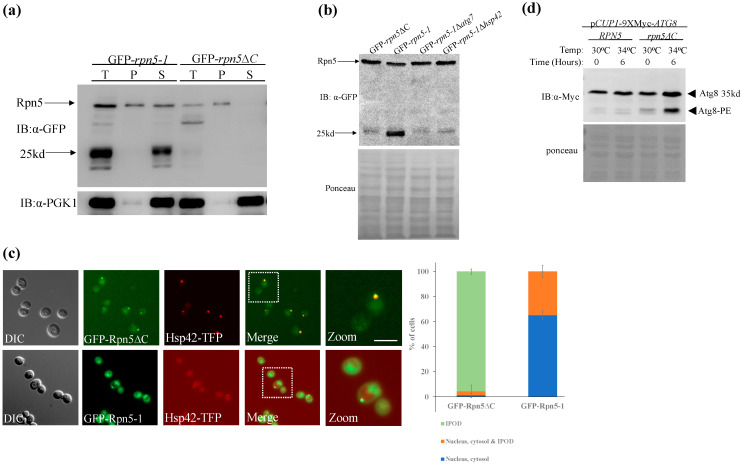
Dysfunctional insoluble proteasomes that accumulate in the IPOD represent an alternative pathway for proteasomes that escape proteaphagy. (**a**,**b**) Cells carrying the ts alleles of *rpn5*ΔC, and *rpn5-1*, expressed under a galactose-inducible promoter (*GAL1*p) (GFP-*rpn5*ΔC and GFP-*rpn5*-1, respectively) were grown in 2% galactose-containing medium to logarithmic phase at the semi-permissive temperature (34 °C), and subjected to the detergent solubility assay. (**b**) Free-GFP signal of *rpn5*-1 overexpressed under a *GAL1* promoter (GFP-*rpn5*-1) is the result of autophagic cleavage. Cells carrying GFP-*rpn5*-1 and deleted in the control autophagic pathway were subjected to free-GFP assay as described in Figure 1. GFP-*rpn5*ΔC was used as a negative control. (**c**) Cell described in A were visualized by differential interference contrast (DIC) microscopy. Images were taken with a 63× objective, and represent a single plane chosen from *z*-series images extending above and below the entire cell. The GFP and mCherry channels were used to visualize the GFP, and TFP fluorescence, respectively. The distribution of GFP-*rpn5*ΔC and GFP-*rpn5*-1 signal was scored as the percentage of cells showing: (i.) Puncta colocalizing with the IPOD marker Hsp42-TFP (IPOD), (ii.) nucleus, cytosol & IPOD (iii.) nucleus, cytosol. Error bars show the standard deviation between three independent experiments (a minimum of 50 cells in each experiment). Bars-5 μm. (**d**) Autophagy is induced in the *rpn5*Δ*C* mutant. Logarithmically growing wt and *rpn5*ΔC cells expressing *ATG8* N-terminally fused to 9xMyc through a copper (CuSo4) inducible promoter (p*CUP1*-9xMyc*-ATG8*) were split to a medium with (+), or without (−) CuSO_4_. Protein extracts were subjected to Western blot analysis and immunoblotted with anti-Myc antibody. Myc-Atg8, and Myc-Atg8 conjugated to the lipid phosphatidylethanolamine (PE), are indicated by black arrows. Ponceau staining was used as the loading control.

**Table 1 biomolecules-13-00077-t001:** Relevant genotype of strains used in this study.

Strain	Genotype	Reference	Substituted/Deleted
BY4741	*MAT*a *his3*Δ*1 leu2*Δ*0 met15*Δ*0 ura3*Δ*0*	[28]	-
BY4742	*MAT*α *his3*Δ*1 leu2*Δ*0 lys2*Δ*0 ura3*Δ*0*	[28]	-
SB148	*MAT*a KmX-*GAL1*-GFP- *rpn5*Δ*c-URA3*	This study	*GAL1*-GFP- *rpn5*Δc
YSB2410	*MAT*a KmX-*GAL1*-GFP-*rpn5-1*	This study	*GAL1*-GFP-*rpn5-1*
YSB2110	*MAT*a *RPN12*-GFP-*HIS3*	This study	*RPN12*-GFP
YSB2154	*MAT*α *RPN12*-GFP-*HIS3 rpn5*Δ*c*	This study	*RPN12, rpn5*Δ*c*
YSB2140	*MAT*α *RPN12*-GFP-*HIS3 rpn5*Δ*c-URA3* Δ*atg7*::HygB	This study	*RPN12*-GFP, *rpn5*Δ*c*, Δ*atg7*
YSB2359	*MAT*α *RPN12*-GFP-*HIS3 rpn5*Δc-*URA3* Δ*pep4*-cloNAT	This study	*RPN12*-GFP, *rpn5*Δ*c*, Δ*pep4*
YSB2171	MATα RPN12-GFP-HIS3 Δatg7::HygB	This study	*RPN12*-GFP, Δ*atg7*
YSB2400	*MAT*a *RPN12*-GFP-*HIS3* Δ*blm10*- KmX	This study	*RPN12*-GFP, Δ*blm10*
YSB2285	*MAT*a *RPN11*-GFP-*HIS3* KmX -*CUP1*-9Xmyc-*ATG8*	This study	*RPN11*-GFP, *CUP1*-9Xmyc-*ATG8*
YSB2286	*MAT*α *RPN11*-GFP-*HIS3* KmX-*CUP1*-9Xmyc-*ATG8 rpn5*Δ*c-URA3*	This study	*RPN11*-GFP, *CUP1*-9Xmyc-*ATG8, rpn5*Δ*c*
YSB2929	*MAT*α *PRE10*-GFP-*HIS3*, *doa5*(ts)-*URA3*	This study	*PRE10*-GFP, *doa5*(ts)
YSB2275	*MAT*α *PRE10*-GFP-*HIS3*	This study	*PRE10*-GFP
YSB2637	*MAT*α *PRE10*-GFP*-HIS3*, Δ*pep4*::cloNAT, *doa5(ts)-URA3*	This study	*PRE10*-GFP, *doa5(ts)*, Δ*pep4*
YSB3410	*MAT*a KmX-*GAL1*-GFP-*rpn5-1* Δ*atg7*::HygB	This study	*GAL1*-GFP*-rpn5-1*, Δ*atg7*
YSB3411	*MAT*a KmX-*GAL1*-GFP-*rpn5-1* Δ*hsp42*::HygB	This study	*GAL1*-GFP*-rpn5-1*, Δ*hsp42*

## Data Availability

Data is contained within the article.

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
