# Peer review of "Inactive Proteasomes Routed to Autophagic Turnover Are Confined within the Soluble Fraction of the Cell"

_biomolecules, 2022, doi:10.3390/biom13010077_

Round 1

Reviewer 1 Report

In the article by Friedman et al. the authors are proposing that proteasomes under nitrogen starvation or dysfunctional proteasomes subjected to autophagy are in a soluble state. The pathway where dysfunctional proteasomes are targeted for autophagy while accumulated in the insoluble fraction takes place mostly as an alternative pathway for proteasomes that escape autophagy upon system saturation and it’s not the rule as currently believed. The authors present well-designed experiments and arguments to substantiate their claims. Nevertheless, there are some points that need to be addressed.

1.     Lines 30-32. Misfolded proteins either manage to refold or they are marked for degradation. The way it is currently written it gives the impression that after refolding the proteins are subjected to degradation. It should be written in a clearer way that it is either the one or the other.

2.     Line 51. It is mentioned that when the proteasome is inactivated the UPS is the preferred pathway. It should be Hsp42-mediated accumulation in the IPOD, which is actually mentioned right afterwards. So this sentence should be removed or the whole part (lines 49-53) should be written in a better way to distinguish what happens when there are still functional or inactive proteasomes.

3.     Can the authors comment on the use of an antibody against PGK1? Is this to show that similar amounts of samples are loaded on the gels? Is there a reason to detect this specific protein?

4.     It would be nice to have an extra column on Table1, where for each strain would be explained what is substituted/deleted relative to the initial wild type strain. This is easier especially for the non-expert that cannot read and understand what each genotype means.

5.     In the first part of the results the authors explain how under nutrient starvation the proteasomes of the soluble fraction are degraded via autophagy. But Figure 1 presents only data for the Rpn12 RP subunit. Are there any data about the CP subunit? Since the authors have a strain (YSB2275) with the Pre10-GFP, it would be interesting to show the fate of the CP subunit under nutrient starvation to complement the findings about the RP subunit.

6.     Deletion of either Atg7 or Pep4 abolishes autophagy. Why do the authors use Atg7 deletion in the first part of the results and Pep4 in the second part? Except there is a specific reason for the change, which should be made clear in the text, it is better to keep a certain consistency between the experiments.

7.     In Figure2a there are extra bands recognized by the anti-GFP antibody in the case of the Pre10. Any idea what they are?

8.     Connected to the previous comment, a very thick extra band is visible for the Rpn12 in Figure2b, which was not visible in Figure2a for the same protein. Can the authors comment on that?

9.     In Figure2c is mentioned that the cells were grown for 3 h, while in the text 4 h are mentioned. Please check it and change accordingly.

Author Response

Reviewer #1:

In the article by Friedman et al. the authors are proposing that proteasomes under nitrogen starvation or dysfunctional proteasomes subjected to autophagy are in a soluble state. The pathway where dysfunctional proteasomes are targeted for autophagy while accumulated in the insoluble fraction takes place mostly as an alternative pathway for proteasomes that escape autophagy upon system saturation and it’s not the rule as currently believed. The authors present well-designed experiments and arguments to substantiate their claims. Nevertheless, there are some points that need to be addressed.

  1. Lines 30-32. Misfolded proteins either manage to refold or they are marked for degradation. The way it is currently written it gives the impression that after refolding the proteins are subjected to degradation. It should be written in a clearer way that it is either the one or the other.

We could not find this sentence in the original version of the article

  1. Line 51. It is mentioned that when the proteasome is inactivated the UPS is the preferred pathway. It should be Hsp42-mediated accumulation in the IPOD, which is actually mentioned right afterwards. So this sentence should be removed or the whole part (lines 49-53) should be written in a better way to distinguish what happens when there are still functional or inactive proteasomes.

We believe that the redundancy between the sentences in lines 49-51 and 51-52 in the original version caused the confusion. Based on this, and the comment made by reviewer #2 we deleted lines 51-52 from the original version.

  1. Can the authors comment on the use of an antibody against PGK1? Is this to show that similar amounts of samples are loaded on the gels? Is there a reason to detect this specific protein?

As mentioned in the legend of Figure 1 in the original version, the antibody against the housekeeping gene 3-phosphoglycerate kinase (anti-PGK1) “served as a loading control, or a to normalize the soluble fraction, respectively”.

  1. It would be nice to have an extra column on Table1, where for each strain would be explained what is substituted/deleted relative to the initial wild type strain. This is easier especially for the non-expert that cannot read and understand what each genotype means.

As suggested by the reviewer, we added an extra column to Table 1 with the requested clarification.

  1. In the first part of the results the authors explain how under nutrient starvation the proteasomes of the soluble fraction are degraded via autophagy. But Figure 1 presents only data for the Rpn12 RP subunit. Are there any data about the CP subunit? Since the authors have a strain (YSB2275) with the Pre10-GFP, it would be interesting to show the fate of the CP subunit under nutrient starvation to complement the findings about the RP subunit.

Our study demonstrated for the first time that the wild-type proteasomes are subjected to autophagy for the soluble fraction of the cells (Figure 1A). The data presented in Figure 1A was shown just to validate previous well-established reports (see paper introduction) showing that wild-type proteasomes are subjected to autophagy under nitrogen starvation.   Since in this process both CP and RP are subjected to vacuolar targeting, we demonstrated the expected degradation occurs using the RP subunit Rpn12. It should be noted that for the second part of our study, proteasome dysfunction was genetically induced using the temperature-sensitive (ts) mutants affecting the RP and CP subunits Rpn5 and Doa5 (rpn5ΔC and doa5-ts respectively). Since dysfunctional CP and RP are independently subjected to proteaphagy, we tested the effect of each ts allele independently.

  1. Deletion of either Atg7 or Pep4 abolishes autophagy. Why do the authors use Atg7 deletion in the first part of the results and Pep4 in the second part? Except there is a specific reason for the change, which should be made clear in the text, it is better to keep a certain consistency between the experiments.

We agree that certain consistency should be maintained between experiments. However, the double Δatg7/rpn5ΔC or Δatg7/doa5-ts mutants showed synthetic growth defects.

As suggested by the reviewer, we added this clarification to the revised version (Lines 217-219).

  1. In Figure 2a there are extra bands recognized by the anti-GFP antibody in the case of the Pre10. Any idea what they are? and 8. Connected to the previous comment, a very thick extra band is visible for the Rpn12 in Figure2b, which was not visible in Figure2a for the same protein. Can the authors comment on that?

 We are not sure what is the nature of these bands. It is possible that in some cases, alongside the linker between GFP and the tagged proteasome subunit another site within the tested proteasome subunit is subjected to vacuolar cleavage, resulting in higher bands.  Unfortunately, we could not identify the specific conditions that induce the appearance of these extra bands. 

  1. In Figure2c is mentioned that the cells were grown for 3 h, while in the text 4 h are mentioned. Please check it and change accordingly.

We thank the reviewer for identifying this mistake.

Corrected to 3 hours in the body of the text.

Reviewer 2 Report

The authors study the autophagy pathway for inactive proteasomes. They provide interesting data on the soluble nature of the proteasome species that can undergo the autophagy pathway. The manuscript is clearly written (with minor exceptions, see below) and the presented data are sound. The overall interpretation given by the authors is also appealing, with particular reference to the major role played by Hsp42 in orchestrating the balance between its aggregase and refoldase activity. I only have a couple of doubts on the interpretation of their data by the authors, and few other minor comments/suggestions.

1) The authors insist on relating the switching on of the IPOD disposal pathway, to the saturation of the autophagy pathway related to soluble species. I am not convinced this interpretation is warranted as is by the reported data. First, the soluble fraction lane for GFP-rpn5ΔC is altogether absent (fig. 3a); yet saturation should not imply the complete switching off of the soluble pathway. Second, most of the signal for the other thermo-sensitive mutant GFP-rpn5-1 is in the soluble fraction lane; why then saturation of the soluble pathway should not occur in this case, when the mutant is as well expressed constitutively? For example, a more reasonable (to me) interpretation would be that the partition between soluble and insoluble fraction is due to other properties that may vary for different mutants, such as expression levels, intrinsic aggregation propensity, fine tuning of the interactions with Hsp42, determining a shift in the balance between the aggregase and the refoldase pathways. Could the authors provide more argument in favor of their statements about saturation of the autophagy machinery?

2) Is it obvious that soluble states are near-native as the authors state more than once? Connected to this point, but non only, I suggest the authors to consider the recent report by Zhu et al (10.1016/j.celrep.2022.111096) where, although in a very different context, on the one hand a similar strategy in analyzing the partition of cell lysate between soluble and pellet fractions is used, and on the other hand emphasis is placed on the misfolded nature of soluble states that may escape the protein quality control system. How would the picture suggested by the authors for inactive proteasome disposal fit with the results by Zhu et al.? At first sight it would look to me an opposite concept: the inactive proteasome needs to be soluble (and near-native?) to interact with the protein quality control machinery, which may add to the interest of the data reported by the authors. I am curious to know what they think about this point.

3) In several cases (fig. 1d, 2a, 2b, 3a) other bands appear in the immunoblots, different from either the freeGFP band and the RPN12 (or rpn5 or PRE10) band. Could the authors mention a possible interpretation for such bands?

4) Page 2, lines 51-52: I find the sentences "As long as functional proteasomes are available, the favored disposal pathway is the degradation of the inactive subunits by the functional 26S complex, similarly to other misfolded complexes. In the case of inactivation of the 26S proteasome, UPS-mediated degradation of its own dysfunctional subunits is the favored disposal pathway" a bit confusing. They seem to repeat the same concept (UPS degrades its own inactive components) although in two opposing cases (26S functional or not).

5) page 4, line 155: the phrasing "on the Δblm10 background" is not clear; I guess the authors mean "in presence of the Δblm10 mutant" or something similar.

6) page 5, line 211: Pup2 is mentioned for the only time (possibly a typo for Pre10?).

7) the authors use a few times (lines 205, 254, 312) the phrasing "terminally" misfolded or aggregated. I am not completely sure what they mean. I would maybe use the "irreversibly" instead.

Author Response

Reviewer #2:

1)              The authors insist on relating the switching on of the IPOD disposal pathway, to the saturation of the autophagy pathway related to soluble species. I am not convinced this interpretation is warranted as is by the reported data. First, the soluble fraction lane for GFP-rpn5ΔC is altogether absent (fig. 3a); yet saturation should not imply the complete switching off of the soluble pathway. Second, most of the signal for the other thermo-sensitive mutant GFP-rpn5-1 is in the soluble fraction lane; why then saturation of the soluble pathway should not occur in this case, when the mutant is as well expressed constitutively? For example, a more reasonable (to me) interpretation would be that the partition between soluble and insoluble fraction is due to other properties that may vary for different mutants, such as expression levels, intrinsic aggregation propensity, fine tuning of the interactions with Hsp42, determining a shift in the balance between the aggregase and the refoldase pathways. Could the authors provide more argument in favor of their statements about saturation of the autophagy machinery?

Previous studies have suggested that the IPOD represents an intermediate compartment before the proteaphagy clearance. However, our findings show that the substrates subjected to proteaphagy are dysfunctional proteasomes in their soluble state, raising the question about the nature of the fraction present in the IPOD. As it is well established that the IPOD represents the final deposition site for amyloidogenic proteins, we suggested that the fraction present in the IPOD represents an alternative pathway for disposal of dysfunctional proteasomes that escaped proteaphagy when the system becomes overloaded.

To address this issue we used the research tools available in our lab, two temperature-sensitive alleles of the lid subunit RPN5, rpn5ΔC, and rpn5-1. These mutants were constitutively expressed to simulate the saturation of the proteaphagy machinery. Overproduced rpn5ΔC was predominantly insoluble and completely confined to the IPOD.  In this case, no free-GFP was detected, most probably because insoluble proteasomes are not subjected to proteaphagy from this fraction. Overproduced rpn5-1 was distributed both in the soluble cytoplasmic fraction and in some cases at the IPOD. In this case, autophagic cleavage was detected solely in the soluble fraction.

We believe that these results support the model that insoluble proteasomes are not the substrate for proteaphagy and that the subunits present in the IPOD represent the dysfunctional proteasomes that escaped proteaphagy.

We completely agree with the reviewer that the different partitioning described for these mutants is the result of their specific nature, and that “saturation should not imply the complete switching-off of the soluble pathway”.  Indeed, rpn5-1 was distributed both in the soluble and insoluble fractions, however, overproduced rpn5ΔC is predominantly insoluble. We can’t rule out that even in this case of rpn5ΔC some fraction is still subjected to proteaphagy. Indeed, when rpn5ΔC was expressed from its endogenous promoter both fractions could be detected (Figure 2b). We believe that the overproduction pushes the balance toward the insoluble state and that the soluble fraction is below the detection level of the methods used.

In light of this comment, this issue was further clarified in the revised version (Lines 281-285)

2)              Is it obvious that soluble states are near-native as the authors state more than once? Connected to this point, but non only, I suggest the authors to consider the recent report by Zhu et al (10.1016/j.celrep.2022.111096) where, although in a very different context, on the one hand a similar strategy in analyzing the partition of cell lysate between soluble and pellet fractions is used, and on the other hand emphasis is placed on the misfolded nature of soluble states that may escape the protein quality control system. How would the picture suggested by the authors for inactive proteasome disposal fit with the results by Zhu et al.? At first sight it would look to me an opposite concept: the inactive proteasome needs to be soluble (and near-native?) to interact with the protein quality control machinery, which may add to the interest of the data reported by the authors. I am curious to know what they think about this point.

Indeed, we believe that proteasomes subjected to proteaphagy are soluble in their near-native state. This issue is extensively discussed in the discussion. In a nutshell, Hsp42 co-aggregates with substrates in a partially unfolded intermediate state, to maintain them in a ready-to-refold conformation

close to the native structure. This co-aggregation is employed to actively control the formation of structures known as CytoQs, and to promote their coalescence into a smaller number of assemblies of larger size, until they are sequestered into the IPOD. Since it is well established that Hsp42 plays an essential role both in CytoQ formation and proteaphagy, we propose that dysfunctional proteasomes are subjected to proteaphagy when embedded in CytoQ in their near-native conformation, and then subjected to autophagic cleavage. When the system becomes saturated, Hsp42 aggregase activity takes over, leading to the accumulation of insoluble proteasomes in the IPOD, which is consistent with its role as the final destination for protein aggregates that cannot be disassembled and could thereby become cytotoxic.

To our understanding the recent report by Zhu et al focuses on a different concept, a subset of newly translated functional proteins that are thermo-sensitive, making them susceptible to misfolding and aggregation only when newly synthesized but not once matured. These proteins are functional and therefore enriched for chaperone binding motifs that assist their proper folding. In our case, we focused on a different unique situation, the required removal of dysfunctional proteasomes for maintaining proteasome homeostasis. We are therefore not sure that it is surprising that the mechanisms are different.

3)              In several cases (fig. 1d, 2a, 2b, 3a) other bands appear in the immunoblots, different from either the freeGFP band and the RPN12 (or rpn5 or PRE10) band. Could the authors mention a possible interpretation for such bands?

We are not sure what is the nature of these bands. It is possible that in some cases, alongside the linker between GFP and the tagged proteasome subunit another site within the tested proteasome subunit is subjected to vacuolar cleavage, resulting in higher bands. Unfortunately, we could not identify the specific conditions that induce the appearance of these extra bands. 

4)              Page 2, lines 51-52: I find the sentences "As long as functional proteasomes are available, the favored disposal pathway is the degradation of the inactive subunits by the functional 26S complex, similarly to other misfolded complexes. In the case of inactivation of the 26S proteasome, UPS-mediated degradation of its own dysfunctional subunits is the favored disposal pathway" a bit confusing. They seem to repeat the same concept (UPS degrades its own inactive components) although in two opposing cases (26S functional or not).

We thank the reviewer for noticing this redundancy. We deleted the second sentence in the revised version.

5)              page 4, line 155: the phrasing "on the Δblm10 background" is not clear; I guess the authors mean "in presence of the Δblm10 mutant" or something similar.

Corrected!

6)              page 5, line 211: Pup2 is mentioned for the only time (possibly a typo for Pre10?).

We thank the reviewer for identifying this mistake.

Corrected!

7)              the authors use a few times (lines 205, 254, 312) the phrasing "terminally" misfolded or aggregated. I am not completely sure what they mean. I would maybe use the "irreversibly" instead.

In light of the reviewer's comment, we replaced “terminally” with "irreversibly"